# ICCONV: A LARGE-SCALE INTENT-ORIENTED AND CONTEXT-AWARE CONVERSATIONAL SEARCH DATASET

## ABSTRACT

In recent years, search engines have made significant advancements. Yet, traditional ad-hoc search engines often struggle with complex search scenarios (e.g. multi-turn information seeking). This challenge has shifted the focus towards conversational search, an approach enabling search engines to interact directly with users to obtain more precise results. Progress in conversational search has been slow due to a lack of data and difficulties in gathering real-world conversational search data. To address these hurdles, we embarked on a journey to autonomously create a large-scale, high-quality conversational search dataset. Previous efforts to create such datasets often overlooked the multi-intent aspect and contextual information, or resulted in a biased dataset, where all dialogue queries linked to a single positive passage. In our study, we have incorporated multi-intent based on the existing search sessions and converted each keyword-based query into multiple natural language queries based on different latent intents present in the related passage. We then contextualized these natural language queries within the same session and organized them into a conversational search tree. A carefully designed dialogue discriminator was utilized to ensure the consistency and coherence of all generated conversations, assessing their quality and filtering out any substandard ones. After extensive data cleaning, we are proud to introduce the **I**ntent-oriented and **C**ontext-aware **Conv**ersational search dataset (ICConv), a large-scale synthetic dataset comprising over 100,000 high-quality, information-seeking conversations. Our human annotators have evaluated ICConv based on six dialogue and search related criteria and it has performed admirably. We further explore the statistical characteristics of ICConv and validate the effectiveness of various conversational search methods using it as a standard for comparison.

## 1 INTRODUCTION

In light of the rapid progression of search engines, users can now effortlessly retrieve routine information using well-formulated keywords (Kobayashi & Takeda, 2000; Liaw & Huang, 2003; Ilan, 1998; Koester, 2006; Purves et al., 2007; Zhan et al., 2021; Gao & Callan, 2021). Nevertheless, traditional ad-hoc search engines, which primarily rely on keyword-based queries, often struggle to capture the user's genuine intent in more complex scenarios. Furthermore, the single-turn interaction of ad-hoc searches frequently delivers a less than ideal user experience, compelling users to incessantly revise their queries to satisfy a sequence of informational demands. These restrictions underscore the pressing need for a comprehensive reinvention of search engine methodologies.

Recently, there has been a surge of interest in conversational search, an interaction framework where users engage with the search engine through multi-turn conversation instead of keywords (Radlinski & Craswell, 2017; Zhang et al., 2018; Rosset et al., 2020; Trippas et al., 2020; Vtyurina et al., 2017; Trippas et al., 2018; Liao et al., 2021; Dubiel et al., 2018). Multi-turn interactive searches cater to a user's shifting information

needs in a more organic manner. The presence of an extensive contextual history also facilitates more accurate and interpretable search outcomes. Thus, conversational search has been viewed as the next-generation information seeking paradigm. However, the frequent appearance of omissions and references in conversations aggravate the complexity of context comprehension. Traditional ad-hoc search techniques and resources may not be suitable for using. One significant hurdle in the evolution of conversational search lies in the scarcity of real-world conversational search data. The acquisition of such data continues to be a substantial impediment. Another challenge is that the smaller artificial datasets like CAsT (Dalton et al., 2020; 2021) do not provide sufficient support for constructing a competent conversational search system. Therefore, synthetic datasets have emerged as a promising solution to the dilemma of data scarcity.

Starting with QuAC (Choi et al., 2018), researchers have developed the conversational search dataset OR-QuAC building upon it (Qu et al., 2020). However, it does not accurately represent real-world scenarios and lacks quality. Subsequently, automated methods were explored for generating conversational search datasets using existing resources and tools. One such method involves converting a large number of web passages into dialogues (Dai et al., 2022), using the sentences within these passages

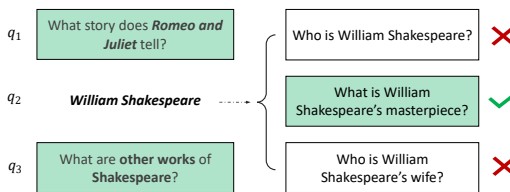

Figure 1: The *multi-intent* phenomenon.

as responses and generating questions based on them. Although this approach can produce a plethora of conversational search data, it suffers from quality issues as the generated questions do not accurately represent the real search intent of users (all questions in a conversation relate have the same positive passage). Another approach attempts to convert existing web search sessions (Mao et al., 2022), which have inherent interaction features and labeled positive samples, into conversations. However, these methods have primarily focused on formally converting in-session keyword queries into in-conversation natural language questions, while neglecting potential contextual features and genuine user intent. But the reality is that a single keyword query usually correspond to multiple natural language queries with different user's intents. This is depicted in Figure 1 as the *multi-intent* phenomenon, which could potentially be resolved by integrating *contextual dependency*.

Based on the findings mentioned above, we strive to automatically construct a high-quality conversational search dataset in this work. Considering positive passages, they often encapsulate many relevant answers (or responses) to a keyword-based query. Each answer potentially mirrors a latent intent of users. We utilize a proficient QA tool (Xiong et al., 2020) to extract evidence for a keyword-based query. The co-supervision of the keyword-based query and its evidence enables us to generate multiple intent-oriented natural language queries. After processing each query in a session, we can construct a query tree where each path signifies a unique de-contextualized search conversation. These generated sessions can be conveniently converted into context-dependent conversations through the process of reverse query reformulation. For conversational consistency and coherence, we further develop a dialogue discriminator to filter the defective conversations. We train it based on contrastive learning (Chuang et al., 2020; Xiao et al., 2020; You et al., 2020; Khosla et al., 2020), whose negative samples are from the destroyed conversations from the real world. Under the multi-intent, our methods traverse all of the possible transitions from the session search data to the conversational search data. With the contextual dependency (Liu et al., 2017; Callejas & Lopez-Cozar, 2008; Ginzburg et al., 1996; Bunt, 1999), we further filter the out-of-context results to guarantee the conversational characteristic of generated conversations. Additionally, we also design the meticulous pipeline to maintain data quality at each step. After implementing our method on MS MARCO search sessions (Nguyen et al., 2016), we develop the **I**ntent-oriented and **C**ontext-aware **Conv**ersational Search Dataset, abbreviated as ICConv. Building upon this, we conduct a human evaluation to assess the quality of our dataset from multiple perspectives, corroborating the high quality of ICConv. Additional statistical experiments are conducted to further explore the underlying properties of our ICConv dataset. Upon reproducing the preceding methods on ICConv, we present an impartial ranking of them and delve into potential explanations.

Our contributions in this work could be summarised as:

- We propose a novel automated data construction method, which considers the multi-intent phenomenon of ad-hoc query and contextual dependency.
- Implementing above method on MS MARCO search sessions, we develop a new large-scale and high-quality conversational search dataset ICConv, relieving the data scarcity problem in this field.
- Through human evaluation and statistical analysis, we analyze the quality and characteristic of ICConv, ensuring that the constructed dataset ICConv is reliable.
- We reproduce the previous conversational search methods on our developed dataset and present a fair ranking of them, as well as analyze thing possible reasons for their ranking.

## 2 RELATED WORK

**TREC CAsT.** TREC Conversational Assistance Track (Dalton et al., 2020; 2021) is a benchmark for evaluating conversational search systems. In order to advance the conversational search progress, this track has presented a new dataset every year since 2019. Though all of the data is collected by humans and has been continually optimized, the scale of the TREC CAST series dataset is too small to support the training of the conversational search model.

**OR-QuAC.** The OR-QuAC (Open-Retrieval Question Answering in Context) dataset is a another benchmark for conversational search (Qu et al., 2020). OR-QuAC consists of questions posed by crowdworkers, along with Wikipedia articles that contain the answers. The collected data have a bias with the real scenario, in which the question of users is always born before the target positive passage. Thus, it is a large-scale but low-quality dataset.

**WikiDialog and WebDialog.** These two datasets are generated by dialogue inpainting (Dai et al., 2022). It views every sentence of a Wikipedia passage as an answer in the context, and then recovers the complete conversation by its question generator. By applying this approach to passages from Wikipedia and the web, they produce WikiDialog and WebDialog, two datasets totaling 19 million diverse information-seeking dialogues. However, all of the queries in the same generated dialogue share a single positive passage, which is a rigorous bias problem.

**ConvTrans.** Another way to automatically construct the conversational search dataset is by utilizing the existing session search log. Search engines produce a large number of search logs every day, which could be organized into sessions based on time. Search sessions have a natural interactive feature, in which users interact with the search engine to seek information. Only to transform keyword-based queries in search sessions into conversational natural language queries, we could yield abundant conversational search data. ConvTrans is an automated dataset constructed by this method (Mao et al., 2022). The negligence of it is that a keyword-based query usually corresponds to multiple natural language queries under different search intents. In addition, the context information is not used well in ConvTrans, causing poor quality of it.

## 3 DATASET CONSTRUCTION

In this section, we provide a comprehensive depiction of the process we implemented to construct the ICConv dataset. (1) We filter out raw search sessions with the potential to be converted into dialogues. (2) Each keyword-based query in the search sessions is expanded into multiple natural language (NL) questions, carefully considering varying user intents. We proceed by structuring each search session into a search tree, where each path symbolizes a rudimentary conversation. (3) We convert these trees into dialogues where the contextualized natural language (CNL) questions incorporate elements of omission, reference, and other

context-dependent features. (4) All generated dialogues are assessed by a discriminator to verify if they align with the standards of a refined conversational format.

## 3.1 SEARCH SESSION FILTERING

We postulate that not all search sessions are apt for conversion into conversations. When the correlation between queries within a session is weak, the ensuing conversation may lack coherence and consistency. To mitigate this issue, we emphasize selecting sessions with robust internal relationships. Specifically, we evaluate the internal connection by gauging the word overlap within a session.

For a search session comprising numerous keyword-based queries, we omit stop words such as 'and', 'a', 'or', etc., and count the instances of query pairs having overlapping words. If the word set of two queries within a session shares **at least one** word in common, we categorize them as a similar pair. We regard a session as having an internal relation if it accommodates **no fewer than two** such pairs, signifying its potential for conversion into a conversation. Upon applying this methodology to MS MARCO search sessions, we find that a mere 13.3% of sessions fulfill the criteria. This implies that most users may not be accustomed to interacting with conventional ad-hoc search systems, thereby underlining the need for the evolution of more interactive conversational search systems.

## 3.2 NL QUESTION GENERATION

Differing from session search, conversational search utilizes NL questions as a more user-friendly approach. Therefore, our primary concern should be how to transform a keyword-based query into an NL question. However, reconstructing an NL question is not a straightforward task due to two reasons: (1) a set of words can correspond to multiple NL questions, and (2) the resulting NL questions are influenced by the user's search intents, which may partially manifest in the expected response.

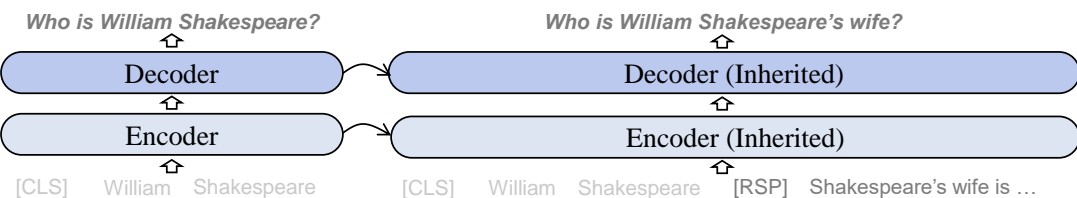

Figure 2: Two-stage method to generate intent-oriented NL questions.

Considering above findings, we propose to reformulate the intent-oriented question with a novel two-step method, which is illustrated in Figure 2. It first converts the keyword-based query into a plain NL question based keyword reconstruction, and then incorporate user's responses to co-generate an intent-oriented NL question.

### 3.2.1 PLAIN NL QUESTION GENERATION

In this section, we utilize the Quora dataset (Aghaebrahimian, 2017), which consists of a large collection of question-and-answer pairs from the Quora website, totaling 400,000 pairs of questions.

To begin, we employ KeyBERT (Grootendorst, 2020) for extracting keywords from the original questions. Our approach is based on the hypothesis that users generally tend to provide detailed ad-hoc queries. Consequently, we set the minimum number of keywords as either $N - 3$ or $0.8 * N$ (where $N$ represents the number of words in the query).

Next, we fine-tune the T5 (Raffel et al., 2020) model to convert the keyword-based queries into NL questions. However, these generated NL questions are generic in nature, as they fail to capture the specific intentions of the users.

### 3.2.2 Intent-oriented NL question generation

For ad-hoc queries, the relevant passages usually reflect more explicit intent. By analyzing these passages, we can gain a better understanding of what users are looking for. However, a single query may be associated with multiple possible answers within a given passage. Taking this into consideration, we systematically generate all possible conversations.

To develop an intent-oriented NL question generator, we build upon the plain NL question generator obtained in the previous step. We train it using two datasets: OR-QuAC (Qu et al., 2020) and QReCC (Anantha et al., 2020), which consist of conversations focused on information needs. Conversations without responses are filtered out. Our approach involves optimizing the maximum likelihood of generating de-contextualized questions when given the pair of keywords extracted from the questions and their corresponding responses, as illustrated in Figure 2.

After training, we extract the potential responses from the positive passages as the users' intent. We utilize a well-trained dense retriever ANCE (Xiong et al., 2020) to identify candidate responses with a matching score above a certain threshold, which is set at 705. Using the extracted candidate responses and their corresponding queries as input, the intent-oriented NL question generator can generate multiple intent-oriented NL questions for each query. This approach enables us to transform each search session into a conversation tree, as depicted in Figure 3. In this tree, each path represents a search conversation, and it is evident that the dark path stands out as the best choice. Furthermore, these intent-oriented NL questions are context-independent and require additional processing to match the contextual context.

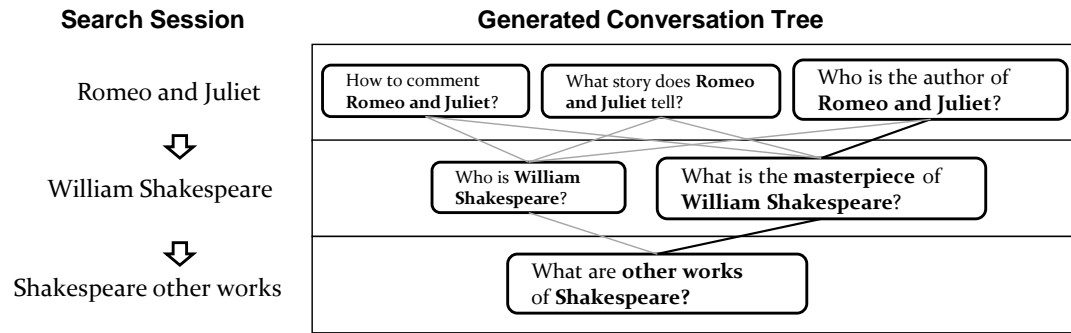

Figure 3: The conversation tree generated by intent-oriented NL question generator. The dark path of the tree means the best conversation.

### 3.3 NL to CNL transformation

In this section, we will focus on how to NL questions into more user-friendly contextualized natural language (CNL) questions. CNL questions serve as an interactive medium that is easier for users to engage with. To accomplish this, we will utilize the query reformulation dataset QReCC (Anantha et al., 2020), which provides pairs of CNL questions and their corresponding de-contextualized NL questions used in conversations.

To train our model, we employed a T5-base model (Raffel et al., 2020) and fine-tuned it to learn the process of transforming NL questions into CNL questions. The process involves providing a de-contextualized NL

question, along with its preceding turns, as input to the model, and expecting it to generate the corresponding CNL question, as shown below:

$$q_k = \text{T5}([q_1, r_1, \ldots, q_{k-1}, r_{k-1}, q_k^*]),$$

where $q_k^*$ is the de-contextual NL question to be transformed, $q_1, r_1, \ldots, q_{k-1}, r_{k-1}$ are the questions and responses in context, $q_k$ is the CNL question. We transform each NL question into a conversation one by one, then a complete search conversation can be obtained. By parsing the entire conversation trees, we totally collect 698,762 search conversations. However, if there are not a further filtering, the most conversations are lack of consistence and coherence, *e.g.,* the light paths in Figure 3. Therefore, in addition to above query-level operation, it is also important to perform dialogue-level operation to disuse the poor conversation.

### 3.4 DIALOGUE QUALITY CONTROLLING

To further enhance the quality of the data generated, we introduce a self-supervised dialogue discriminator (depicted in Figure 4) that assesses the completeness of a dialogue. We utilize BERT (Devlin et al., 2018) as the underlying model and employ contrastive learning to fine-tune it. We employ a data manipulation technique that disrupts the dialogue structure by randomly deleting, inserting, or replacing utterance to generate negative samples.

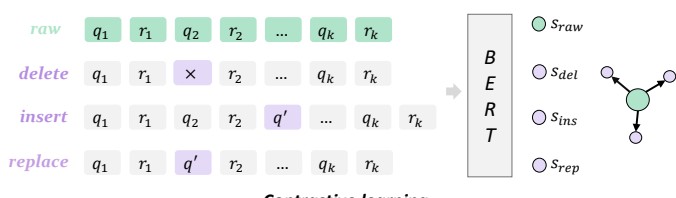

Figure 4: The self-supervised dialogue quality discriminator.

Subsequently, we optimize the model using contrastive loss with the negative samples.

After the training phase, we employ the dialogue discriminator to evaluate all the generated sessions and exclude those that exhibit weak coherence and consistency. This process yields 105,811 high-quality conversations, accounting for 15.2% of the total sessions. Subsequently, we divide all these conversations into training, development, and test sets in an 8:1:1 ratio.

## 4 DATASET ANALYSIS

### 4.1 BASIC STATISTICS.

The basic statistical results of ICConv, as shown in Table 1. Overall, the statistis exhibit a high degree of consistency in the distribution of each subset. With over 100,000 dialogues and 700,000 questions, it can be considered a large-scale conversational search dataset.

Table 1: The statistics of splitted ICConv.

| Statistics | Train | Dev | Test |
|---|---|---|---|
| # Conversation | 84,704 | 10,589 | 10,588 |
| # Question | 585,129 | 73,371 | 73,569 |
| # Avg. Token / Conversation | 186.41 | 187.20 | 187.55 |
| # Avg. Token / Question | 5.50 | 5.50 | 5.51 |
| # Avg. Token / Passage | 58.03 | 58.24 | 57.66 |

## 4.2 TURNS DISTRIBUTION.

The distribution of turns of ICConvis as Figure 5. we observed that the turns distribution in each subset exhibits a consistent, monotonically decreasing trend, with longer conversations having a smaller ratio in ICConv, indicating that users typically prefer to interact with the system for a moderate number of turns. Although not shown in the Figure 5, the maximum turns of ICConvcan reach 73, which is a challenge to long conversation modeling.

## 4.3 QUESTION TYPES.

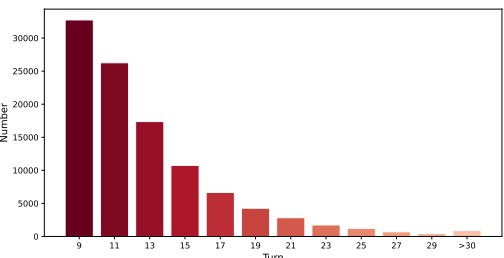

Figure 5: Distribution of turns in ICConv.

We examine the types of questions generated in IC-Conv. The distinguishing characteristic of natural language questions is typically found in the first few words. To analyze the data, we aggregate all the questions from each subset of ICConv and count the number of 1-term and 2-term starting words. To simplify the data, we remove the long tail start words and only reserve the top-8. The resulting statistics are presented in Figure 6. The outer ring of the pie chart shows that over 80% of the questions begin with the word "what", indicating that ICConv predominantly focuses on factual content and requires contextual understanding abilities from conversational search models. The second most common 1-term start word is "how", suggesting that methods or manners are also frequently queried. Although other question types take a small proportion, diverse question types are also be referred, *e.g.,* 'is' (asking for factual correctness), 'where' (asking for place), 'who' (asking for people), etc. From the inner ring, the diversity of question type seems also apparent. These different type questions indirectly verify the effectiveness of our automated construction method, and show the high quality of ICConv.

## 5 HUMAN EVALUATION

We randomly selected 1000 samples and invited annotators to assess the quality of ICConv using six questions[1]. **Q1-Information**: Is the question information-seeking? **Q2-Consistency**: How relevant is the question to the conversation? **Q3-Diversity**: How specific is the question? **Q4-Relevance**: How relevant is the question to the response? **Q5-Correctness**: How grammatically correct is the question? **Q6-Coherence**: How coherent is the question with its context?

The evaluation results are illustrated in Figure 7. Almost all of the questions in the three subsets were information-seeking questions, indicating that our automated dataset construction method has explicit search intent. The generated questions displayed a high level of relevance to the conversations, showing that contextual relevance has been well modeled into ICConv. However, due to material constraints (MS search sessions), the generated questions often ask for factual information, resulting

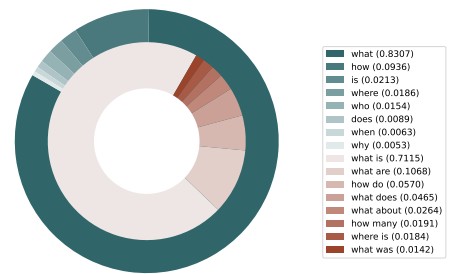

Figure 6: The start word distribution of generated questions. We report the top-8 results of 1-term (outer ring) and 2-term(inner ring). The decimals in the legend are the ratio of start words.

---

[1]Higher scores indicate better performance. Please refer to the appendix for details of the human evaluation.

in relatively poor specificity for ICConv. It is worth noting that the questions in ICConv are closely related to their responses, indicating the effectiveness of our intent-oriented question generation method. Since the responses are selected from positive passages, the relevance between the questions and their responses is nearly equivalent to the relevance between the questions and their positive passages. This ensures the reliability of ICConv as a retrieval dataset. We also assessed the correctness of the generated questions and achieved favorable results, thanks to our meticulous data manipulation pipeline, which filters out most of the unreadable generated content. Lastly, we examined the coherence of the questions in conversations, which is a high-level and abstract characteristic of the conversations. It can be observed that, with the assistance of our dialogue discriminator, most questions in the conversations demonstrate a certain level of coherence, indicating the high quality of ICConv. Based on the human evaluation results across multiple criteria, ICConv exhibits high readability and reliability.

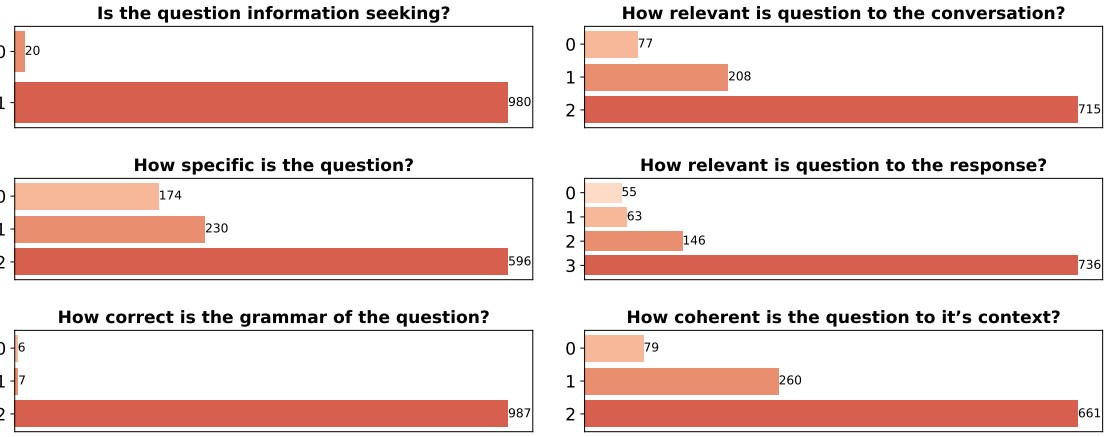

Figure 7: Human evaluation results for ICConv.

# 6 MODEL EVALUATION

In this section. we reproduce previous representative conversational search methods and evaluate their performance. Based on the evaluation result, we will analyze the effectiveness of different methods and introduce our findings.

## 6.1 EXPERIMENTAL SETTING

We compare three taxonomy of existing conversational search methods, which are ad-hoc search, query rewriting and conversational dense retrieval methods.

**Ad-hoc retriever.** We simply do search by exploiting a sparse retriever `BM25` Robertson et al. (2009) and a dense retriever `ANCE` Xiong et al. (2020) with user's last query.

**Query rewriting.** Earlier works focus on the *two-stage* methods, where a delicately devised query rewriter is firstly used to reformulate the context-dependency query to de-contextualized query. Then the latter is used to retrieve by a ad-hoc retriver (here we use `ANCE` uniformly) at second stage. We choose `QuReTeC-QR` Vaku-lenko et al. (2021), which add missing context from the conversation history to the context-dependency query, `GPT2-QR` Radford et al. (2019), and a our implemented `T5-QR`. Besides, we also report the `Manual` results,

where the de-contextualized query is used to perform ad-hoc search. The `Manual` is supposed to the upper bound of all the two-stage methods.

**Conversational dense retriever.** Recently a few works start to encode the whole context to avoid information vanishing. Hence, they are more effecient as the *one-stage* method. Here we compare the `ConvDR` Yu et al. (2020), which adopts a teacher-student framework to distill the dense representation of reformulation query to a conversational dense retriever, `ContQE` Lin et al. (2021), which construct a dataset with pseudo-relevance labels to train the retriever, and `ConvEnc`, a conversational encoder implemented by us.

All of above models are evaluated by the MRR, NDCG@3, NDCG@10, Recall@5, Recall@20 and Recall@100 metrics.

## 6.2 EXPERIMENTAL RESULTS

**Overall Performance.** Overall, our evaluation of various models on the ICConv dataset indicates poor performance due to the unique manner in which the data was constructed. Specifically, intent-oriented natural language question generation and context-aware dialogue discriminator were used to generate and filter data based on session search from real-world scenarios, making retrieval difficult for traditional conversation search models. Among the models tested, `ConvDR` achieved the best performance, while `BM25` performed the worst. This is not surprising, as traditional sparse retrieval methods are ill-suited for conversational search, whereas `ConvDR` is customized for this task.

Table 2: The performance of compared methods on ICConv. The best results are in bold face. ♡ and ♠ denote two-stage and one-stage methods respectively. ◇ denote ad-hoc search methods.

| | MRR | NDCG@3 | NDCG@10 | Recall@5 | Recall@20 | Recall@100 |
|---|---|---|---|---|---|---|
| BM25$^\diamond$ | 0.0611 | 0.0436 | 0.0716 | 0.0892 | 0.2078 | 0.4283 |
| ANCE$^\diamond$ | 0.1741 | 0.1413 | 0.2105 | 0.2547 | 0.5190 | 0.7710 |
| GPT2-QR$^\heartsuit$ | 0.1743 | 0.1413 | 0.2108 | 0.2554 | 0.5199 | 0.7721 |
| QuReTeC-QR$^\heartsuit$ | 0.1797 | 0.1433 | 0.2176 | 0.2675 | 0.5383 | 0.8048 |
| T5-QR$^\heartsuit$ | 0.1950 | 0.1579 | 0.2359 | 0.2868 | 0.5769 | 0.8535 |
| Manual$^\heartsuit$ | 0.2035 | 0.1657 | 0.2462 | 0.2993 | 0.5979 | 0.8782 |
| ContQE$^\spadesuit$ | 0.1046 | 0.0835 | 0.1227 | 0.1510 | 0.3083 | 0.5249 |
| ConvEnc$^\spadesuit$ | 0.1981 | 0.1627 | 0.2431 | 0.3061 | 0.5761 | 0.7877 |
| ConvDR$^\spadesuit$ | **0.2601** | **0.2272** | **0.3139** | **0.3994** | **0.6659** | **0.8666** |

**Sparse vs. Dense.** Based on the comparison between sparse retrieval methods such as the `BM25` and dense retrieval methods like the `ANCE` series, it is clear that the latter outperforms the former in terms of performance. One potential reason for this discrepancy is that understanding natural language questions requires the consideration of more semantic information. `ANCE` is trained using hard negative samples, which enhances the encoder's representational capabilities, while `BM25` only focuses on token-level information. Especially in conversational search scenario, only token-level information is hard to express the real intent of users.

**Rewrite vs. Manual.** we found that the performance of `T5-QR` surpasses all of the rewrite methods. The encoder-decoder architecture may obtain more effective contextual information than decoder-only architecture and improve the co-reference resolution. Nevertheless, there is also a performance gap between these rewrite methods and manual rewriting, indicating the query resolution still deserve to research and how to probe more implicit contextual information to help query resolution is still a challenge.

**Two-stage vs. One-stage.** The two-stage method of query reformulation involves converting a contextualized query into a de-contextualized one before directly using it for ad-hoc retrieval. However, this method discards valuable contextual information, which can lead to imprecise question comprehension. In contrast, the one-stage method directly encodes the entire conversation into a dense vector, making it a more efficient and effective approach. It is noteworthy that most one-stage methods outperform two-stage methods, with the exception of `ContQE`. We believe that this exception is due to its pseudo-label construction mechanism, which is used to address data scarcity in conversational search but unfortunately introduces noise and reduces the retrieval accuracy in ICConv. Specifically, the `ConvDR` keeps ahead among all the methods and is obviously better than the last two conversational dense retriever, which show that both the knowledge distillation loss (which focuses on context understanding) and the ranking loss (which focuses on question-passage matching) improve the model's performance. This inspires us to consider more latent information from both the dialogue aspect and the search aspect, which may help improve the conversational search system.

Taking into account the findings above, we suggest that conversational dense retrieval (a one-stage and dense method) is a more suitable approach for conversational search. We also encourage the consideration of potential information from both dialogue and search to improve the conversational search system. We believe that the release of the ICConv will present a new challenge for conversational search as well as advance progress in this field.

## 7 CONCLUSION

In conclusion, this paper presents the Automated Intent-oriented and Context-aware Conversational Search dataset (ICConv), a large-scale and high-quality dataset that enables the development of more accurate and effective conversational search systems. Previous attempts to create conversational search systems have been hindered by a lack of data, resulting in poor performance. We present a novel automated dataset construction method to overcome the challenge of data scarcity by considering multi-intent and contextual information based on the existing search sessions. Implementing it on the MS MARCO search session, we build the automated dataset ICConv. ICConv is evaluated by human annotators and performs well on six criteria related to dialogues and search. Additionally, we conduct an analysis of the statistical characteristics of ICConv and evaluate the performance of various conversational search models on the dataset to ensure a fair comparison. We suppose that the large-scale and high-quality ICConv dataset has the potential to advance the field of conversational search and improve the accuracy of search results.

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

## A    EVALUATION CRITERIA

To evaluate the quality of ICConv, we invited several annotators to assess the generated questions based on multiple dialogue principles as follows.

**Is the question information-seeking?** In conversational search, questions are intended to seek information. To assess the retrieval ability of conversational search models accurately, we aim to minimize the proportion of open-domain questions in our dataset. Choose either "Yes (1)" or "No (0)."

**How relevant is the question to the conversation?** A question in a conversation should be related to the context of the discussion. Abrupt questions can impact the realism of the conversation. Choose one of the following options: "Not at all (0)" (the question is completely irrelevant to its context), "Relevant topic (1)" (the question's topic aligns with its context), or "Follows up (2)" (the question should be understood within the conversation's context).

**How specific is the question?** To evaluate the diversity of generated questions in ICConv, annotators rate the specificity of the questions. Choose from three options: "Not at all (0)", "Somewhat (1)", or "Very (2)."

**How relevant is the question to the response?** This criterion assesses the correspondence between the question's intent and the response. As mentioned earlier, we consider the user intent reflected in the response from the positive passage when generating the search conversation. High relevance between the question and the response indicates the effectiveness of our method. Choose one of the following: "Not at all (0)", "Incompletely (1)" (the question only partially refers to the response, or vice versa), "Sufficiently (2)" (the question largely refers to the response, or vice versa), or "Perfectly (3)".

**How correct is the grammar of the question?** A qualified natural language dataset should adhere to strict grammatical requirements to ensure readability. Choose from the following options: "Not at all (0)", "Somewhat (1)," or "Very (2)".

**How coherent is the question with its context?** This criterion measures the logical relevance between the question and the context, rather than the content relevance. Choose one of the following options: "Not at all (0)", "Somewhat (1)", or "Very (2)".

# B CASE STUDY

*What foods contain b12?*

There are many vegan foods fortified with B12.

*Does the supplement have any side effects?*

Based on studies involving individuals, it appears 1 mg of cyanocobalamin via injection does not create any notable side effects.

*How much do i take daily?*

1 For the general adult population, a daily dose of the smallest available tablet of B12 (usually 100 mcg) should be sufficient.

*What is b complex vitamins?*

B-complex vitamins are a mixture of eight essential B-vitamins that our bodies require on a daily basis.

*What types of fortified cereal are there?*

Wheaties is made by General Mills and is a fortified cereal.

*What food is vitamin b12 in?*

Vitamin B12 is naturally found in animal products, including fish, meat, poultry, eggs, milk, and milk products.

Figure 8: An example in ICConv datset. The light red rounded rectangles mean generated questions based on session query while the light blue ones mean responses selected from the positive passages.

We have chosen an example generated by ICConv, as shown in Figure 8. From a micro perspective, the generated questions are diverse and mostly relevant to the responses, and they are well-organized in a logical order. For instance, they start with fact-oriented questions such as "Which foods contain b12?" and gradually transition to manner-oriented questions like "How much b12 should I take daily?". It seems to simulate a patient to continuously learn more about the knowledge of supplements. From a macro perspective, the conversation is well-integrated and coherent, making it difficult to determine if it was constructed automatically or in the real world. Therefore, we believe that our intent-oriented and context-aware method played a crucial role in creating such a flawless conversation in ICConv.

## C    ICCONV DATASHEET

The original questions are in **bold**. The subtext to each question is in *italics*. The answers are in plain text with no formatting.[2]

### C.1    MOTIVATION

The questions in this section are primarily intended to encourage dataset creators to clearly articulate their reasons for creating the dataset and to promote transparency about funding interests.

**For what purpose was the dataset created?** *Was there a specific task in mind? Was there a specific gap that needed to be filled? Please provide a description.*

We constructed this dataset to relieve the data scarcity problem of conversational search to an extent. Data scarcity has been a problem hindering the research of conversational search. Due to the high cost of manual construction, some methods tried to build datasets automatically. However, existing methods for building conversational search datasets either overlook the multi-intent problem and contextual information or create a biased dataset where all queries in a conversation are related to a single positive passage. Considering the multi-intent problem and contextual information, we constructed this large-scale intent-oriented and context-aware dataset automatically based on the web search session data in MS MARCO [3].

**Who created the dataset (e.g., which team, research group) and on behalf of which entity (e.g., company, institution, organization)?**

ICConv research group.

**Who funded the creation of the dataset?** *If there is an associated grant, please provide the name of the grantor and the grant name and number.*

No.

**Any other comments?**

No.

### C.2    COMPOSITION

Most of these questions are intended to provide dataset consumers with the information they need to make informed decisions about using the dataset for specific tasks. The answers to some of these questions reveal information about compliance with the EU's General Data Protection Regulation (GDPR) or comparable regulations in other jurisdictions.

---

[2]The questions were copied from the paper Datasheets for Datasets https://arxiv.org/pdf/1803.09010.pdf

[3]The link to MS MARCO is https://microsoft.github.io/msmarco. It is MIT-licensed.

**What do the instances that comprise the dataset represent (e.g., documents, photos, people, countries)?** *Are there multiple types of instances (e.g., movies, users, and ratings; people and interactions between them; nodes and edges)? Please provide a description.*

Conversations converted from web search sessions, and positive passages related to the queries in conversations.

**How many instances are there in total (of each type, if appropriate)?**

105,881 conversational search sessions in total.

**Does the dataset contain all possible instances or is it a sample (not necessarily random) of instances from a larger set?** *If the dataset is a sample, then what is the larger set? Is the sample representative of the larger set (e.g., geographic coverage)? If so, please describe how this representativeness was validated/verified. If it is not representative of the larger set, please describe why not (e.g., to cover a more diverse range of instances, because instances were withheld or unavailable).*

The material of this dataset is a sample from a larger set. The larger set is MS MARCO search sessions. We filtered these search sessions and reserved a subset of them. We converted the subset into our dataset. The subset contains the sessions which have more potential for conversion into a conversation.

**What data does each instance consist of?** *"Raw" data (e.g., unprocessed text or images) or features? In either case, please provide a description.*

Each instance, i.e., conversation, consists of multi-turn conversational queries, oracle queries, responses, and positive passages related to queries.

**Is there a label or target associated with each instance?** *If so, please provide a description.*

Each query in the conversations is associated with a positive passage.

**Is any information missing from individual instances?** *If so, please provide a description, explaining why this information is missing (e.g., because it was unavailable). This does not include intentionally removed information, but might include, e.g., redacted text.*

No.

**Are relationships between individual instances made explicit (e.g., users' movie ratings, social network links)?** *If so, please describe how these relationships are made explicit.*

No.

**Are there recommended data splits (e.g., training, development/validation, testing)?** *If so, please provide a description of these splits, explaining the rationale behind them.*

We have splitted the data into train, dev, and test sets at a ratio of 8:1:1.

**Are there any errors, sources of noise, or redundancies in the dataset?** *If so, please provide a description.*

Errors may exist. The sessions in MS MARCO we chose to reserve may have potential errors, which may still exist in our dataset.

There are two sources of noise. On the one hand, our data was generated based on MS MARCO, which may has potential noise. On the other hand, although we have implemented quality control measures, the generation process is inherently uncontrollable to some extent, which inevitably introduces noise into our data.

No redundancies. Because every conversation was converted from a unique session.

**Is the dataset self-contained, or does it link to or otherwise rely on external resources (e.g., websites, tweets, other datasets)?** *If it links to or relies on external resources, a) are there guarantees that they will exist, and remain constant, over time; b) are there official archival versions of the complete dataset (i.e., including the external resources as they existed at the time the dataset was created); c) are there any restrictions (e.g., licenses, fees) associated with any of the external resources that might apply to a dataset consumer? Please provide descriptions of all external resources and any restrictions associated with them, as well as links or other access points, as appropriate.*

It is self-contained.

**Does the dataset contain data that might be considered confidential (e.g., data that is protected by legal privilege or by doctor-patient confidentiality, data that includes the content of individuals' non-public communications)?** *If so, please provide a description.*

No.

**Does the dataset contain data that, if viewed directly, might be offensive, insulting, threatening, or might otherwise cause anxiety?** *If so, please describe why.*

No.

**Does the dataset identify any subpopulations (e.g., by age, gender)?** *If so, please describe how these subpopulations are identified and provide a description of their respective distributions within the dataset.*

No.

**Is it possible to identify individuals (i.e., one or more natural persons), either directly or indirectly (i.e., in combination with other data) from the dataset?** *If so, please describe how.*

No.

**Does the dataset contain data that might be considered sensitive in any way (e.g., data that reveals race or ethnic origins, sexual orientations, religious beliefs, political opinions or union memberships, or locations; financial or health data; biometric or genetic data; forms of government identification, such as social security numbers; criminal history)?** *If so, please provide a description.*

No.

**Any other comments?**

No.

### C.3 COLLECTION PROCESS

As with the questions in the previous section, dataset creators should read through these questions prior to any data collection to flag potential issues and then provide answers once collection is complete. In addition to the goals outlined in the previous section, the questions in this section are designed to elicit information that may help researchers and practitioners to create alternative datasets with similar characteristics. Again, questions that apply only to datasets that relate to people are grouped together at the end of the section.

**How was the data associated with each instance acquired?** *Was the data directly observable (e.g., raw text, movie ratings), reported by subjects (e.g., survey responses), or indirectly inferred/derived from other data (e.g., part-of-speech tags, model-based guesses for age or language)? If the data was reported by subjects or indirectly inferred/derived from other data, was the data validated/verified? If so, please describe how.*

Every conversation was automatically converted from a web search session in MS MARCO.

**What mechanisms or procedures were used to collect the data (e.g., hardware apparatuses or sensors, manual human curation, software programs, software APIs)?** *How were these mechanisms or procedures validated?*

We design a complex pipeline to convert the raw search sessions into high-quality search conversations. The details of the technique have been reported in our paper.

**If the dataset is a sample from a larger set, what was the sampling strategy (e.g., deterministic, probabilistic with specific sampling probabilities)?**

We suppose that not all search sessions are suitable for conversion into conversations. When the relationship between queries in a session is weak, the resulting conversation may lack coherence and consistency. To address this issue, we focus on selecting sessions with stronger internal relations as our material. Specifically, we assess the internal relation by analyzing the word overlap within a session. Given a search session with several keyword-based queries, we remove stop words such as "and," "a," and "or", etc., and calculate the number of query pairs that have overlapping words. If the word set of two queries in a session has at least one common word, we treat them to as a similar pair. We consider a session to have an internal relation if it contains no fewer than two similar pairs, indicating its potential for conversion into a conversation. After filtering, only 13.3% of MS MARCO search sessions are reserved.

**Who was involved in the data collection process (e.g., students, crowdworkers, contractors) and how were they compensated (e.g., how much were crowdworkers paid)?**

The data of ICConv is automatically generated without manual collection.

**Over what timeframe was the data collected?** *Does this timeframe match the creation timeframe of the data associated with the instances (e.g., recent crawl of old news articles)? If not, please describe the timeframe in which the data associated with the instances was created.*

MS MARCO web search sessions dataset was sampled from Bing usage logs from 2018-06-01 to 2018-11-30. ICConv was built on it.

**Were any ethical review processes conducted (e.g., by an institutional review board)?** *If so, please provide a description of these review processes, including the outcomes, as well as a link or other access point to any supporting documentation.*

No.

**Did you collect the data from the individuals in question directly, or obtain it via third parties or other sources (e.g., websites)?**

We generated the data automatically based on MS MARCO.

**Were the individuals in question notified about the data collection?** *If so, please describe (or show with screenshots or other information) how notice was provided, and provide a link or other access point to, or otherwise reproduce, the exact language of the notification itself.*

No. No individual is in question. MS MARCO is open source.

**Did the individuals in question consent to the collection and use of their data?** *If so, please describe (or show with screenshots or other information) how consent was requested and provided, and provide a link or other access point to, or otherwise reproduce, the exact language to which the individuals consented.*

N/A.

**If consent was obtained, were the consenting individuals provided with a mechanism to revoke their consent in the future or for certain uses?** *If so, please provide a description, as well as a link or other access point to the mechanism (if appropriate).*

N/A.

**Has an analysis of the potential impact of the dataset and its use on data subjects (e.g., a data protection impact analysis) been conducted?** *If so, please provide a description of this analysis, including the outcomes, as well as a link or other access point to any supporting documentation.*

No.

**Any other comments?**

No.

### C.4 PREPROCESSING/CLEANING/LABELING

Dataset creators should read through these questions prior to any preprocessing, cleaning, or labeling and then provide answers once these tasks are complete. The questions in this section are intended to provide dataset consumers with the information they need to determine whether the "raw" data has been processed in ways that are compatible with their chosen tasks. For example, text that has been converted into a "bag-of-words" is not suitable for tasks involving word order.

**Was any preprocessing/cleaning/labeling of the data done (e.g., discretization or bucketing, tokenization, part-of-speech tagging, SIFT feature extraction, removal of instances, processing of missing values)?** *If so, please provide a description. If not, you may skip the remaining questions in this section.*

We processed the web search session data from MS MARCO. The processing includes filtering, converting keywords-based queries to natural-language queries, converting natural-language queries into conversational natural-language queries, and quality controlling. After that, no preprocessing/cleaning/labeling was done to the converted data.

**Was the "raw" data saved in addition to the preprocessed/cleaned/labeled data (e.g., to support unanticipated future uses)?** *If so, please provide a link or other access point to the "raw" data.*

N/A.

**Is the software that was used to preprocess/clean/label the data available?** *If so, please provide a link or other access point.*

N/A.

**Any other comments?**

No.

### C.5 USES

The questions in this section are intended to encourage dataset creators to reflect on the tasks for which the dataset should and should not be used. By explicitly highlighting these tasks, dataset creators can help dataset consumers to make informed decisions, thereby avoiding potential risks or harms.

**Has the dataset been used for any tasks already?** *If so, please provide a description.*

Yes. We used this dataset to evaluate existing methods, including BM25, ANCE, ConvDR, and ContQE as well as several variants of them. We described this in section 6 of our paper.

**Is there a repository that links to any or all papers or systems that use the dataset?** *If so, please provide a link or other access point.*

No. We did not track the papers and systems that use our dataset.

**What (other) tasks could the dataset be used for?**

The dataset could be used for other information-seeking conversation tasks like conversational question answering.

**Is there anything about the composition of the dataset or the way it was collected and preprocessed/-cleaned/labeled that might impact future uses?** *For example, is there anything that a dataset consumer might need to know to avoid uses that could result in unfair treatment of individuals or groups (e.g., stereotyping, quality of service issues) or other risks or harms (e.g., legal risks, financial harms)? If so, please provide a description. Is there anything a dataset consumer could do to mitigate these risks or harms?*

No.

**Are there tasks for which the dataset should not be used?** *If so, please provide a description.*

This dataset is not intended to be used in a task that would cause or is likely to cause overall harm.

**Any other comments?**

No.

## C.6 DISTRIBUTION

Dataset creators should provide answers to these questions prior to distributing the dataset either internally within the entity on behalf of which the dataset was created or externally to third parties.

**Will the dataset be distributed to third parties outside of the entity (e.g., company, institution, organization) on behalf of which the dataset was created?** *If so, please provide a description.*

No.

**How will the dataset will be distributed (e.g., tarball on website, API, GitHub)?** *Does the dataset have a digital object identifier (DOI)?*

We have released the dataset through GitHub and the repository link is here. No DOI.

**When will the dataset be distributed?**

It has been distributed.

**Will the dataset be distributed under a copyright or other intellectual property (IP) license, and/or under applicable terms of use (ToU)?** *If so, please describe this license and/or ToU, and provide a link or other access point to, or otherwise reproduce, any relevant licensing terms or ToU, as well as any fees associated with these restrictions.*

The dataset is distributed under CC BY-SA 4.0 license[4].

**Have any third parties imposed IP-based or other restrictions on the data associated with the instances?** *If so, please describe these restrictions, and provide a link or other access point to, or otherwise reproduce, any relevant licensing terms, as well as any fees associated with these restrictions.*

No.

---

[4]https://creativecommons.org/licenses/by-sa/4.0/legalcode

**Do any export controls or other regulatory restrictions apply to the dataset or to individual instances?** *If so, please describe these restrictions, and provide a link or other access point to, or otherwise reproduce, any supporting documentation.*

No.

**Any other comments?**

No.

### C.7 MAINTENANCE

As with the questions in the previous section, dataset creators should provide answers to these questions prior to distributing the dataset. The questions in this section are intended to encourage dataset creators to plan for dataset maintenance and communicate this plan to dataset consumers.

**Who will be supporting/hosting/maintaining the dataset? How can the owner/curator/manager of the dataset be contacted (e.g., email address)? Is there an erratum?** *If so, please provide a link or other access point.*

The ICConv research group handles the hosting and maintenance. We host the dataset here. The manager can be contacted through email: quantu@ruc.edu.cn. We will put the updating information on GitHub.

**Will the dataset be updated (e.g., to correct labeling errors, add new instances, delete instances)?** *If so, please describe how often, by whom, and how updates will be communicated to dataset consumers (e.g., mailing list, GitHub)?*

No. The dataset is static, but we will fix the errors and update it on GitHub.

**If the dataset relates to people, are there applicable limits on the retention of the data associated with the instances (e.g., were the individuals in question told that their data would be retained for a fixed period of time and then deleted)?** *If so, please describe these limits and explain how they will be enforced.*

No.

**Will older versions of the dataset continue to be supported/hosted/maintained?** *If so, please describe how. If not, please describe how its obsolescence will be communicated to dataset consumers.*

Yes. The static dataset will be hosted and maintained on GitHub.

**If others want to extend/augment/build on/contribute to the dataset, is there a mechanism for them to do so?** *If so, please provide a description. Will these contributions be validated/verified? If so, please describe how. If not, why not? Is there a process for communicating/distributing these contributions to dataset consumers? If so, please provide a description.*

Our dataset is open source under MIT license. People who want to contribute to the dataset can contact us or create issues on GitHub. And people who want to extend/augment/build on the dataset can re-distribute under the same license.

**Any other comments?**

No.

# D    DATA ACCESS

We uploaded the tarball of the dataset to GitHub. Everyone can download it from this URL: https://github.com/hongjx175/ICConv, or you can use git (lfs) to clone and pull the repository.

# E    STATEMENT OF RESPONSIBILITY

The authors declare that they bear all responsibility for violations of rights related to this dataset and that it is MIT-licensed.

# F    HOSTING AND MAINTENANCE PLAN

We host and maintain the dataset on GitHub, and the link is here. ICConv is a static dataset. We will fix the identified errors.

# G    META DATA

The data is in JSON format. The dataset contains three JSON files: `train_sessions.json`, `dev_sessions.json`, `test_sessions.json`. Each JSON file contains a list of JSON objects, where each JSON object represents a dialogue.

The following are the fields of JSON:

- *session_id* (string): The unique identifier of the conversation reserved from MS MARCO.
- *turns* (list of JSON objects): A list of the conversation turns. The fields in every turn are:
    - *qid* (int): The original qid of the web search query used to convert in MS MARCO.
    - *query* (string): The conversational natural language query.
    - *oracle_query* (string): The natural language query, which is de-contextualized.
    - *answer* (string): The response to the query we extracted from the positive passage.
    - *passage* (int & string): The ID of the positive passage and its content. The passages are from MS MARCO passage V1.

Here is an example:

```json
{
    "session_id": "marco-gen-dev-761127",
    "turns": [
        {
            "qid": 566556,
            "query": "What are the symptoms of bronchitis?",
            "oracle_query": "What are the symptoms of bronchitis?",
            "answer": "Symptoms of bronchitis include coughing up
                yellow-grey mucus, sore throat, wheezing and having a
                blocked nose.",
            "passage": [
                1511891,
```

```
                    "Cold \& flu health centre. Bronchitis. Bronchitis is a
                        common infection causing inflammation and
                        irritation to the main airways of the lungs.
                        Symptoms of bronchitis include coughing up yellow-
                        grey mucus, sore throat, wheezing and having a
                        blocked nose. Acute bronchitis may be responsible
                        for the hacking cough and phlegm production that
                        sometimes accompany an upper respiratory infection.
                        In most cases, the infection is viral in origin, but
                        sometimes it's caused by bacteria."
            ]
        },
        {
            "qid": 784788,
            "query": "What is pneumonia?",
            "oracle_query": "What is pneumonia?",
            "answer": "Pneumonia (nu-MO-ne-ah) is an infection in one
                or both of the lungs.",
            "passage": [
                1011041,
                "Pneumonia (nu-MO-ne-ah) is an infection in one or both
                    of the lungs. Many germs-such as bacteria, viruses,
                    and fungi-can cause pneumonia.The infection
                    inflames your lungs' air sacs, which are called
                    alveoli (al-VEE-uhl-eye).neumonia (nu-MO-ne-ah) is
                    an infection in one or both of the lungs. Many germs
                    -such as bacteria, viruses, and fungi-can cause
                    pneumonia."
            ]
        },
        {
            "qid": 476203,
            "query": "What are its symptoms?",
            "oracle_query": "What are pneumonia's symptoms?",
            "answer": "Symptoms also can vary, depending on whether
                your pneumonia is bacterial or viral.",
            "passage": [
                385922,
                "Symptoms also can vary, depending on whether your
                    pneumonia is bacterial or viral. 1  In bacterial
                    pneumonia, your temperature may rise as high as 105
                    degrees F. 2  The initial symptoms of viral
                    pneumonia are the same as influenza symptoms: fever,
                    a dry cough, headache, muscle pain, and weakness."
            ]
        },
        {
            "qid": 508239,
            "query": "What are the symptoms of mono in adults?",
            "oracle_query": "What are the symptoms of mono in adults?",
```

```json
            "answer": "2  The symptoms of mono include: 3  fever, 4
                fatigue, 5  sore throat, and.",
            "passage": [
                1184563,
                "1 Most adults have laboratory evidence (antibodies
                    against the EBV) indicative of a previous infection
                    with EBV and are immune to further infection. 2  The
                     symptoms of mono include: 3  fever, 4  fatigue, 5
                    sore throat, and. 6  swollen lymph nodes. 7  The
                    diagnosis of mono is confirmed by blood tests."
            ]
        },
        {
            "qid": 507997,
            "query": "What about an inner ear infection?",
            "oracle_query": "What are the symptoms of an inner ear
                infection?",
            "answer": "Symptoms include dizziness, loss of balance,
                nausea, vomiting, tinnitus, and vertigo.",
            "passage": [
                1150712,
                "Labyrinthitis is an inner ear disorder. It occurs when
                     a vestibular nerve, important to spatial navigation
                     and balance control, becomes inflamed. Symptoms
                    include dizziness, loss of balance, nausea, vomiting
                    , tinnitus, and vertigo. With proper treatment, most
                     people find relief from symptoms within 1 to 3
                    weeks."
            ]
        }
    ]
}
```

## H  DISCUSSION ON PREVIOUS REVIEWERS' CONCERNS

In our previous submission to SIGIR 2023 Resource Track, despite unanimous agreement from all reviewers on the merits of our work, it is regrettable that we neglected to include a documentation explaining the data. As a result, our submission was rejected. However, this time, we have provided detailed supplementary materials and a documentation to facilitate the utilization of our dataset in future research.

