# OpenReview forum: "ICConv: A Large-Scale Intent-Oriented and Context-Aware Conversational Search Dataset"
_ICLR.cc/2025/Conference — Submitted to ICLR 2025_

### Official Review · Reviewer_GT9o · 2024-11-03

**Soundness:** 2
**Presentation:** 1
**Contribution:** 3
**Rating:** 3
**Confidence:** 4

**Summary:**

This paper introduces a conversational search dataset, ICConv and its construction method, highlighting the novel approach of considering multiple intents within a single keyword-based query. The overall methodology to construct the dataset can be divided into 4 stages. 1) Filter MS Marco search sessions with the number of overlapping words. 2) Generate intent-oriented natural language questions by giving a keyword and a retrieved response as a input to fine-tuned T5. 3) Transform each natural language question into contextualized natural language question. 4) Filter out dialogues by evaluating coherence and consistency. In addition, authors analyzed the dataset and provided the results from human evaluations along with the performance of conversational search methods on ICConv.

**Strengths:**

1. **Well-motivated and timely work on conversational search:** it is evident that crawling the users’ log to aggregate conversation sessions to construct a conversational dataset have issues on privacy, thus there is a substantial need to automatically construct the conversation search dataset in large scale. In this sense, authors point out that limitations of prior works and proposed to resolve the problem.
2. **Supplementary materials:** Authors provided useful supplementary materials to provide examples and the github link to the dataset which helps readers to understand the structure and contents of the dataset.

**Weaknesses:**

1. **Lack of demonstration of details on constructing dataset:** The writing is enough to understand the overall procedure, but the details (either in words or formulas) or justification of the method design is poor. For example, in section 3.1, the complete list of stop words can be included either in the main body or in the appendix. Also in section 3.1, authors claim that having few instances (probably from MS MARCO Conversational Search DEV dataset while it is written as MS MARCO search sessions) after filtering with overlapping keywords implies that users are not used to interacting with conventional search systems. But it is not sure that conventional systems hinder the users’ usage of overlapping keywords in conversational manner, because there is a case that users might have complex information needs that cannot be captured in lexical manner but can be captured with semantic similarities. Furthermore, in section 3.4, the logical flow from training BERT with contrastive loss to finding the optimal path between all possible sessions with filtering out weak coherence is not straightforward and lacks of details. In conclusion, it would be helpful if authors provide the figure of the overall method, the table with comparison to other datasets and articulate texts.

2. **Comparison to prior works with regards to novelty:** Authors point out that ConvTrans neglected that “keyword-based query usually corresponds to multiple natural language queries under different intents”, which is important claim to support the significance of the proposed method. Both ConvTrans and ICConv are generated by extending keywords to NL and CNL questions with T5. While the key novelty of the proposed method is generating multi-intent questions via concatenating responses as inputs (section 3.2.2) and filtering the path (section 3.4), the novelty of the other components is minimal.

**Questions:**

1. Please upload the dataset in an anonymous repository for the fair review
2. Are keywords (extracted in section 3.2.1) used to generate NL queries one-by-one? If so, are there overlapping keywords within a session?
3. Could you please show the distribution of the number of conversation paths derived from each MS Marco session?
4. What is the number of annotators in section 5 and how did authors recruit them?
5. Are you certain there are no more references for the method design (section 3), excluding the dataset and model?
6. Multi-intent phenomenon is the problem when the dataset is naively constructed from a keyword-based queries. Considering this feature can amplify the size of dataset as authors highlighted. Besides, what is the key difference from ICConv to CAsT or ConvTrans in terms of contents and quality of conversations?

---

### Official Review · Reviewer_Z4np · 2024-11-05

**Soundness:** 2
**Presentation:** 3
**Contribution:** 2
**Rating:** 3
**Confidence:** 4

**Summary:**

This paper addresses the scenarios requiring multi-turn and context-aware interactions, where the development is hindered by a lack of high-quality, real-world conversational search data. To tackle this, the paper introduces ICConv, a new large-scale dataset specifically designed for conversational search. ICConv is built on MS MARCO search sessions and addresses the multi-intent phenomenon—where a single keyword query can represent multiple underlying intents.

**Strengths:**

The resource has the potential to be impactful since it is designed for multi-turn scenarios, where a common problem is the lack of data.

**Weaknesses:**

My main concern regards the difficult of this dataset when using LLMs. While retrieval is a challenging part, I struggle to understand how this dataset can pose new challenges for the community - questions are from MSMARCO and are potentially easy to answer, even in a multi-turn scenario. The paper would benefit from an end-to-end evaluation where it can be shown that better retrieval and maybe better RAG models are required. Also, findings on question rewriting seems to be in line with a previous work in the field (Question rewriting for open-domain conversational qa: Best practices and limitations) that in depth analyzed the impact of QR. I suggest the authors to strengthen the justification part of this work, carefully describe differences with previous work, and clearly articulate (with experiments) the new challenges that this dataset introduces.

**Questions:**

see my comments above

---

### Official Review · Reviewer_rwtC · 2024-11-05

**Soundness:** 3
**Presentation:** 3
**Contribution:** 2
**Rating:** 5
**Confidence:** 3

**Summary:**

This paper presents a new conversational search dataset which is a synthetic dialogue dataset consisting of ~100000 information-seeking dialogues based on MS MARCO search sessions. It thoroughly describes the data construction method to achieve the ultimate high-quality conversational search dataset using NLP models. A thorough statistical analysis of the dataset is provided and a human evaluation demonstrates the high quality of the new dataset. Previous conversational search models are evaluated on the dataset, followed by a logical analysis of their performance.

**Strengths:**

1. The paper is well-written and easy to follow.
2. The data construction methodology is valid, which may provide insights into other data collection tasks.
3. A human evaluation corroborates the quality of the dataset and benchmarks are provided for researchers to advance their research in this area.

**Weaknesses:**

1. The paper's contribution would be more solid if a new conversational search model that outperforms the previous methods on the new dataset was also proposed along with its release.
2. LLMs have shown remarkable capabilities in text generation. Why are they not used in the data construction process or data evaluation to further enhance the dataset quality?

**Questions:**

Please refer to the weakness part.

---

### Official Review · Reviewer_j1ur · 2024-11-05

**Soundness:** 1
**Presentation:** 1
**Contribution:** 1
**Rating:** 3
**Confidence:** 5

**Summary:**

This paper presents ICConv, a large-scale conversational search dataset comprising over 100k conversations generated through an automated pipeline. The dataset construction process consists of 4 key stages: session filtering based on word overlap, question generation that captures diverse user intents, query contextualization, and quality control using a dialogue discriminator. This work is trying to address two critical gaps in existing datasets: accommodating multiple user intents and effectively integrating contextual dependencies across dialogue turns. Authors conduct evaluations including both human assessment and statistical analysis to position ICConv as a robust benchmark for evaluating a range of conversational search methods.

**Strengths:**

1) This paper is constructing an automatic workflow to generate conversational datasets to reduce human efforts while balancing quality.
2) The size of this dataset is large-scale.
2) Authors implemented two kinds of mainstream technique for conversational searching evaluation: Query Rewriting and End-to-end.
3) Authors analyze and propose hypothesis about their experiments to inspire readers to understand difficulties of conversational search.

**Weaknesses:**

The paper addresses an important problem in automatically constructing realistic and complex datasets for conversational search. However, the work would benefit from significant revisions, particularly in clarifying the motivation, improving writing clarity, and ensuring a thorough understanding of experimental design.

## Motivation Deservers more clear and detailed illustrations:
1) The introduction falls short in clarity, particularly regarding the claim in Lines 38-39 that traditional keyword-based search engines struggle to capture **"genuine"** user intents. This claim is not well supported and could be more convincingly illustrated. From a practical standpoint, many users including me successfully find relevant information through keyword-based searches, which suggests keyword searching still occupies a central role in realistic search interactions. A clearer definition and distinction between "genuine" and "non-genuine" queries, supported by references, would strengthen this argument. Additionally, a figure comparing genuine and non-genuine queries could help clarify this distinction. The authors should also elaborate on why previous single-turn methods and datasets fail to capture these genuine intents, and how a multi-turn conversational search engine could potentially address these limitations. Moreover, the premise that single-turn queries deliver a less than ideal user experience seems weak. If users want to refine their searches, they can rephrase their queries or conduct new searches. Figure 15 also seems more aligned with a dialogue-based QA system, like ChatGPT, rather than a "realistic" conversational search engine, which typically ranks results (e.g., Google Search) not a single answer. Therefore, the initial premise should be reconsidered or better justified to avoid overstating the limitations of single-turn search.

2) Several sentences require more precise expressions, making it difficult to follow the motivation. For instance, "Traditional ad-hoc search techniques and resources may not be suitable for using" is vague. What is and specifically makes them **unsuitable**? If the term "complexity" is key here, it should be explicitly defined (does it refer to longer query history, more heterogeneous contexts, or more flexible query expressions?) Providing more citations and fine-grained explanations would help make the motivation clearer and more compelling.

3) While the paper emphasizes the **"real-world"** aspect of the proposed dataset, this concept remains unclear. What specific features make this dataset more "real-world" than prior datasets? The authors use terms like "genuine" and "complex" but without clear definitions or examples. Including concrete examples, illustrations, or comparisons would help define these terms and demonstrate the advantages of this dataset over existing ones.

4) The explanation of "multi-intent queries" in Lines 71-73 is difficult to follow. An one-sentence definition would help clarify what is meant by "multi-intent" or "single-intent." Does this imply that a single query could have multiple relevant documents or answers? Furthermore, Figure 1 is overly simplistic and lacks sufficient captions to convey the intended message. It’s unclear where "single-intent" and "multi-intent" distinctions are made in this figure, and the signal of "NO" in the first and third queries is confusing. Revising the figure to include more general domain examples would make it more accessible, particularly for readers who may not be familiar with the Shakespearean content used here.

5) The paper references MS MARCO abruptly without providing an introduction or context. A brief introduction of what is MS MARCO, along and why it was chosen as the primary corpus in this paper, would improve clarity for readers unfamiliar with this resource.

6) Others:
   - In Line 46, why do authors switch to "interactive" from "conversational"? Do they have the same meanings in this project? According to recent research [1-3], the "interactive" mainly refers to the interaction with the environment but for single-turn queries.
   - In Line 46, what does "organic manner" mean?

## No Task Definitions and Preliminaries:
I suggest adding a specific section on task definitions and preliminaries to improve clarity. Without this, readers may find it challenging to follow the terminology. For example, terms like "session," "query tree," "conversation tree," and "search tree" should be clearly introduced. Are these distinct concepts, or do they refer to similar structures? Additionally, it would be helpful to specify how many tree algorithms are implemented and the rationale behind their selection. A section that defines key concepts, such as "turn," "user query," "session," "tree," and "ground truth answers", which would provide readers with a solid foundation for understanding the paper.

I also find it challenging to understand the task output and the evaluation methods. The authors should clearly introduce the metrics used to evaluate model performance for this task. Currently, there appears to be an inconsistency: **Figure 15 suggests the task output resembles a QA or dialogue task (with a single natural language sentence as the ground truth answer)**, whereas the results in Table 2 are evaluated using ranking- and recall-based metrics. How are metrics like recall or MRR computed with a natural language sentence as the answer? Did authors implement keyword retrieval from GT NL answers? A detailed description of the task formulation, evaluation metrics, ground truth answers, and evaluation methodology is **essential**.

## Methodology
1) No Comparison with Related Works: The authors do not provide a statistical comparison between the features of their dataset and those of other conversational search datasets. Such a comparison, highlighting comprehensive features, would help clarify the unique contributions of this dataset. In Lines 175-176, the authors mention that their method is "novel." However, they should explicitly explain how it differs from previous approaches to better illustrate this novelty. For instance, how is this work different from recent research [4], which also trains smaller models to simulate user-engine interactions? The authors should clarify their distinct contributions and novel aspects, especially since [4] appears to employ more rigorous workflows and metrics for quality control.

2) Potential Bias in Filtering: The filtering strategy, which relies on counting overlapping words to construct pairs, may introduce bias. This approach seems insufficiently rigorous; for instance, it might not accurately capture queries with **negation**, such as "I don’t like lobster" or "I need recipes without lobster." These queries could still include overlapping words and lead to irrelevant results (e.g., recommendations or posts about "lobster"). How do the authors address such cases? Moreover, simple word overlap may not fully capture the nuanced and varied expressions of user queries, especially if the dataset aims to reflect "real-world" interactions.

3) In Section 3.4, the negative samples are generated through rule-based methods. However, this approach may be insufficient for capturing the complexities of real-world interactions. Relying on rule-based operations could oversimplify the task, potentially making the training target too easy and leading to models that fail to distinguish between grammatically correct but contextually incoherent conversations. A more detailed explanation of how these rules align with real-world conversational features would strengthen this section.

### Human Evaluation

1) **Unrealistic Number of Turns**: The number of turns in some conversations is very high (e.g., 73 turns), which seems unrealistic, as users are unlikely to maintain this level of engagement in real-world settings. If authors deem that it's realistic, please provide references or reports to prove this.

2) **Biased Benchmark**: Sections 4.3 and 5 suggest that the benchmark, generated entirely through automated methods, may contain inherent biases:
   - In Figure 6, the majority of questions begin with "what," indicating an uneven distribution of question types.
   - In Line 326, the authors acknowledge that the method may suffer from limited diversity. However, diversity is quite important for a benchmark or dataset.

3) **Inconsistencies in Human Evaluation Metrics**:
   - The distinctions between certain evaluation questions are unclear. For instance, Q2 and Q6 appear similar; can the authors clarify the difference between these questions?
   - Q3 is also ambiguous, as the connection between "specificity" and "diversity" is not immediately clear.
   - Additionally, the term "question" is used inconsistently. The authors frequently refer to the "diversity of questions" and the "grammatical correctness of questions." Are these questions referring to the final turn, the initial turn, or all turns in the conversation? More specific guidelines on how to evaluate each question (Q2, Q3, etc.) and criteria for assigning ratings (1, 2, or 3) would be helpful.
   - The rating scale in Figure 7 lacks consistency. Some items have four rating levels, while others have three. What is the reason for this variation? Furthermore, the results seem to indicate that overall quality is not particularly high as highlighted in Introduction.

### Questionable Experimental Conclusions

1) **Unclear Definition of "Manual" Results**: It is unclear what is meant by "Manual" results in Line 375. Further clarification of this term and its relevance would be helpful.

2) **Ambiguous Conclusion on Model Comparison**: In Lines 419-420, the authors conclude that "encoder-decoder models perform better than decoder-only models." However, this statement lacks depth and rigorous analysis. For instance, could this difference be due to variations in model parameter sizes? Llama-3-8B, which is also a decoder-only model, could serve as a comparison point. It would be beneficial for the authors to substantiate this claim by comparing models of similar parameter sizes, such as T5-3B and Llama, to better illustrate the impact of architecture rather than parameter count.

3) **Questionable Conclusions on "Two-Stage vs. One-Stage" Approaches**:
   - **Implementing One-Stage for Long Dialogs**: How do the authors handle the one-stage approach in T5-3B with a token limit of 512 for lengthy dialogs? Given that some dialogs include lengthy history such as 73 turns, token limits may lead to issues with long-context handling. Additional explanation on this implementation would clarify the comparison. It seems authors did some process for it since it may lead to 512 / 73 = 8 tokens for each question and response in the history on average. According to the example shown in Figure 19, it obviously not true.
   - **Two-Stage Implementation for Query Rewriting**: It is unclear how the two-stage process is implemented for query rewriting. Does this involve recursively rewriting queries or loading and rewriting all queries at once? This detail is crucial for understanding the approach's effectiveness.
   - **Potential Bias in Results**: In Line 418, the authors observe that T5-QR outperforms other models. Could this exceptional performance be influenced by implementing T5 as the backbone for both data generation for benchmarking? This could introduce a bias in the evaluation. The authors should analyze and provide evidence to show that such bias is not present since this is a significant issue when using model-generated data for benchmarking purposes.

Overall, the authors omit critical implementation details regarding model configurations. They should provide more information, such as methods for handling long-context inputs and specific approaches for query rewriting.



[1] WebArena: A Realistic Web Environment for Building Autonomous Agents \
[2] InterCode: Standardizing and Benchmarking Interactive Coding with Execution Feedback \
[3] OSWorld: Benchmarking Multimodal Agents for Open-Ended Tasks in Real Computer Environments \
[4] Learning to Simulate Natural Language Feedback for Interactive Semantic Parsing

**Questions:**

See details in Weakness, and I summarized and listed more crucial questions here:

1. What specifically defines "genuine" user intents, as mentioned in Lines 38-39? How do these differ from intents captured by traditional keyword-based search engines?

2. How do multi-turn conversational search engines improve user experience compared to single-turn search? Why can’t users simply refine their queries in a single-turn search? How do the authors define "multi-intent" versus "single-intent" queries? Does this imply that one query can yield multiple relevant documents, or that queries are inherently ambiguous?

3. What characteristics make this dataset more "real-world" than existing datasets? Can the authors provide examples or features to illustrate this claim? And compare your datasets with previous works by detailed statistical analysis.

4. **Terminology Definitions in Task Setup**: Can the authors define terms like "session," "query tree," "conversation tree," and "search tree"? Are these distinct concepts, or do they overlap? How many tree algorithms are implemented, and what is their purpose? Add a task formulation or definition before using many terms.

5. How are recall and MRR computed given that the ground truth is in natural language (NL) sentence format? Is there a retrieval step for keywords, or another approach? Additionally, why do Figure 15 and Table 2 suggest different evaluation outputs?

6. Given the reliance on overlapping words, how do the authors handle negations or flexible expressions just by counting overlapping words? (e.g., "I need recipes without lobster") to ensure relevant query pairs? Could this approach introduce unintended biases?

7. For the T5-3B model, which has a token limit of 512, how do the authors handle lengthy dialogs (e.g., 73 turns) within the one-stage approach? Was any context truncation or special processing applied?

8. Could the higher performance of T5-QR be influenced by the use of T5 as both a data generator and benchmark model? Have the authors analyzed this possible bias, and if so, what evidence demonstrates that the evaluation is unbiased?

---

### Official Review · Reviewer_YECM · 2024-11-07

**Soundness:** 3
**Presentation:** 2
**Contribution:** 3
**Rating:** 5
**Confidence:** 4

**Summary:**

The work proposes an intent oriented, multi-turn, context-aware conversational search dataset. The method proposed to construct this synthetic dataset is novel considering multi-intent nature and contextual information for formulating natural language queries. The automated data construction approach also develops a dialogue discriminator model to control for dialogue quality and evaluates existing systems

**Strengths:**

1. The authors tackle an important problem of bridging the data scarcity gap in conversational search with multiple intents and contextual awareness.

2. The data construction method is novel and the manual and automated evaluations are comprehensive.

3. The data has utility to the IR and NLP communities with applications being conversational QA, search.

**Weaknesses:**

1. Several choices are not clearly explained or supported. For instance, grouping queries within a session that share at least one common word is not well supported as it could result in false positives and false negatives. For instance, queries like “American president” and “leader of United states” may not be grouped. Additionally queries like “american election” and “american universities” may be grouped which reflect two very different intents resulting in false positives. Additionally it is not clear why ANCE was chosen to select candidate responses for aiding in generating questions and the threshold mechanism is also not clearly specified. It is also clear in cases where there are no responses that meet the threshold are the queries ignored ? If so, how is the completeness of the conversation maintained by the proposed approach ? Additionally, the question generation approach does not explicitly constrain for the metrics measured, such as coherence and completeness of the conversation. For instance, the approach of followup question generation explicitly tries to optimize for completeness. In this approach, an initial query with response could be used to generate a NL query and the response to this question could be leveraged to generate the followup question. As mentioned earlier grouping session queries only based on lexical match with an arbitrary heuristic as done in ICCONV does not guarantee they are related and hence might result in conversational turns in dataset that are bit incoherent. While the example in Appendix is coherent, on closer look at the dataset released, i observed certain inconsistencies where a conversational turn were on unrelated topics. For instance, marco-gen-train-7146805 in the dev set starts with query “What is google classroom?” and the followup queries in turn are “What is facilitated diffusion?” which is a huge drift in topic and intent with no clear connection. Similarly “marco-gen-train-7746920” consists of queries about medications which are in no way related to another and not representative of real-world conversational search interactions which is one of the main motivations of this work.


2. Some key details are missing in the work. What dataset was used to train the dialogue discriminator ? What was the retrieval corpus used for the experiments on icconv ? was it the original corpus for MSMARCO ? Or only the positive passages used to generate ICCONV ? I think this is critical to mention this information clearly as the corpus should also reflect real-world scenario of open-domain search with presence of distractors reflecting real-world challenges for retrievers.


3. Also some key related works are missing. For instance [1] is a very relevant recent dataset on conversational search with information seeking queries and is critical to compare and distinguish ICConv contributions to this work. Likewise, ConvSDG[2] also is relevant. Also the baseline comparisons are bit outdated and some relevant works especially query reformulation approaches for conversational search such as ConvGQR[3] which is quite recent and ConQRR[4] are relevant to be included for comparison on the curated benchmark for conversational search.



[1] ProMISe: A Proactive Multi-turn Dialogue Dataset for Information-seeking Intent Resolution Yash Butala, Siddhant Garg, Pratyay Banerjee, Amita Misra

[2] ConvSDG: Session Data Generation for Conversational Search, Fengran et. al

[3] ConvGQR: Generative Query Reformulation for Conversational Search Fengran Mo, Kelong Mao, Yutao Zhu, Yihong Wu, Kaiyu Huang, Jian-Yun Nie

[4] CONQRR: Conversational Query Rewriting for Retrieval with Reinforcement Learning Zeqiu Wu, Yi Luan, Hannah Rashkin, David Reitter, Hannaneh Hajishirzi, Mari Ostendorf, Gaurav Singh Tomar

**Questions:**

1. Why do you think that grouping session queries only based on lexical match makes sense ?  In my opinion  an arbitrary heuristic as done in ICCONV does not guarantee they are related and hence might result in conversational turns in dataset that are incoherent.

2. While the example in Appendix is coherent, on closer look at the dataset released, i observed certain inconsistencies where a conversational turn were on unrelated topics. For instance, marco-gen-train-7146805 in the dev set starts with query “What is google classroom?” and the followup queries in turn are “What is facilitated diffusion?” which is a huge drift in topic and intent with no clear connection. Similarly “marco-gen-train-7746920” consists of queries about medications which are in no way related to another and not representative of real-world conversational search interactions which is one of the main motivations of this work.

3. What dataset was used to train the dialogue discriminator ?

4. What was the retrieval corpus used for the experiments on icconv ? was it the original corpus for MSMARCO ? Or only the positive passages used to generate ICCONV ?

---

### Official Review · Reviewer_ZP27 · 2024-11-09

**Soundness:** 2
**Presentation:** 3
**Contribution:** 2
**Rating:** 5
**Confidence:** 3

**Summary:**

This paper describes ICConv, a new synthetic multi-turn conversational search dataset. The dataset is prepared by selecting a relevant subset of MS MARCO search sessions and expanding the keyword-based searches present in MS MARCO into context-aware natural language questions that could formulate a multi-turn dialogue. The authors provide summary statistics and human validation (on a subset) of the dataset to demonstrate its quality. Further, the authors evaluate search methods which follow different paradigms (ad-hoc, query rewriting-based, and dense conversational) to compare and contrast different methods on the task proposed in ICConv.

**Strengths:**

* The paper is generally well-written and easy to follow. It describes the dataset construction process in significant detail.
* Given that the conversational/chat-based mechanisms are becoming a more common modality of interaction, the proposed dataset fills an important niche.
* The dataset is documented extensively through a datasheet provided in the appendix.

**Weaknesses:**

*  Missing rationale for choices made (in terms of models/thresholds chosen) during the dataset preparation phase.
* Recent baselines and experimental setup details missing from the comparisons provided experimental results section.
* Inconsistent scaling for human evaluation experiments.

**Questions:**

* Please provide the rationale for following choices made during the dataset preparation phase:
	* Choosing the number of keywords as $N-3$ or $0.8 * N$
	* Choosing the dense retriever (ANCE) threshold of 705 during NL question generation
	* Choosing T5/BERT models, and the setup (training/hyperparameters etc) used for question generation/dialog quality control, respectively
* The most recent included baseline is ContQE by Lin et al (2021). Why are more recent methods such as COTED (Mao et al, 2022) or ConvGQR (Mo et al, 2023) excluded?
* Why are different scales used across the questions asked in the human evaluation?

Mao, Kelong, et al. "Curriculum contrastive context denoising for few-shot conversational dense retrieval." SIGIR 2022.

Mo, Fengran, et al. "ConvGQR: Generative Query Reformulation for Conversational Search.", ACL 2023.

Minor comments:
* Line 051: Traditional ad-hoc search ... using -> this sentence is unclear, needs to be rephrased.
* Line 087: design ~~the~~ a meticulous
* Line 113: is ~~a~~ another benchmark
* Line 366: taxonomy is the incorrect nomenclature here, maybe paradigm?

**Details Of Ethics Concerns:**

Public github repository link (non-anonymized) shared in the paper (Appendix D, Data Access section)

---

### Meta-Review · Area_Chair_3SjB · 2024-12-19

**Metareview:**

- Scientific Claims and Findings:
    - This paper presents a new synthetic multi-turn conversational search dataset and its construction method. It includes summary statistics, human validation on a subset of the dataset to demonstrate its quality, and an evaluation of various conversational search models on the dataset.
- Strengths:
   - The dataset may be valuable to IR and NLP community.
   - The dataset construction method provides a reliable automated workflow.
- Weaknesses:
    - Many important details about dataset construction and evaluation are missing or unclear.
    - The justification for this work, including comparison to prior works in terms of novelty, and the new challenges introduced by this dataset, is not clearly described.
- Most Important Reasons for Decision:
     - Given the identified weaknesses, the work is not yet ready for publication at this conference.

**Additional Comments On Reviewer Discussion:**

All six reviewers unanimously rated the paper below the acceptance threshold. The authors did not provide a rebuttal, and there was no discussion between the authors and reviewers.

---

### Decision · Program_Chairs · 2025-01-22

Reject